# Application of a Deep Learning Approach to Analyze Large-Scale MRI Data of the Spine

**DOI:** 10.3390/healthcare10112132

**Published:** 2022-10-26

**Authors:** Felix Streckenbach, Gundram Leifert, Thomas Beyer, Anita Mesanovic, Hanna Wäscher, Daniel Cantré, Sönke Langner, Marc-André Weber, Tobias Lindner

**Affiliations:** 1Department of Diagnostic and Interventional Radiology, Pediatric Radiology and Neuroradiology, University Medical Center Rostock, 18057 Rostock, Germany; 2Core Facility Multimodal Small Animal Imaging, Rostock University Medical Center, 18057 Rostock, Germany

**Keywords:** German National Cohort, MRI, spine, artificial intelligence, large-scale data, convolutional neural network, normative data

## Abstract

With its standardized MRI datasets of the entire spine, the German National Cohort (GNC) has the potential to deliver standardized biometric reference values for intervertebral discs (VD), vertebral bodies (VB) and spinal canal (SC). To handle such large-scale big data, artificial intelligence (AI) tools are needed. In this manuscript, we will present an AI software tool to analyze spine MRI and generate normative standard values. 330 representative GNC MRI datasets were randomly selected in equal distribution regarding parameters of age, sex and height. By using a 3D U-Net, an AI algorithm was trained, validated and tested. Finally, the machine learning algorithm explored the full dataset (*n* = 10,215). VB, VD and SC were successfully segmented and analyzed by using an AI-based algorithm. A software tool was developed to analyze spine-MRI and provide age, sex, and height-matched comparative biometric data. Using an AI algorithm, the reliable segmentation of MRI datasets of the entire spine from the GNC was possible and achieved an excellent agreement with manually segmented datasets. With the analysis of the total GNC MRI dataset with almost 30,000 subjects, it will be possible to generate real normative standard values in the future.

## 1. Introduction

Computer-aided tools with implemented artificial intelligence (AI) algorithms constitute an exciting and growing field with solutions for medicine as well. These solutions provide support for diagnosis and treatment planning for intervention in many clinical areas. In the field of radiology, AI applications are conceivable for almost any work task. In addition, many solutions for typical radiological tasks have already been brought into the area [1,2,3,4,5,6,7,8].

In the last years, the number of magnetic resonance imaging (MRI) examinations has increased substantially, especially spine MRI in the field of musculoskeletal and neurological applications, for example, in patients with chronic back pain [9]. In Germany, there has been an increase of 71% in MRI examinations between 2007 and 2016, but only a growth of 33% in radiologists within the last ten years [10,11,12].

The aim of the German National Cohort (GNC), a large population-based MR study with over 200,000 people from Germany, is to investigate the causes of the development of major chronic diseases. The analysis of the participant’s individual anatomy is an essential part of the study and supports the correct diagnosis and treatment planning [13].

Degenerative disc disease (DDD) is a common imaging finding in the clinical routine. It may be present with or without symptoms, and it is usually detected when symptoms occur. Although there are a variety of grading scales to categorize the imaging findings [14], little is known regarding the extent to which DDD is physiological in a certain population, depending on their age and gender. Currently, there are large-scale-based quantitative biomarkers and data for morphometric analysis of the spine in relation to age, sex and height available [15,16] but no real standardized reference values [17]. In these studies, the spine was just partially analyzed, or segmentation was performed just manually. In addition, the data of these studies were always acquired on a geographically local basis, like in the “Study of Health in Pomerania” [15,16]. To our best knowledge, until now, there is no study that evaluated the morphometry of the entire spine of more than 11.000 subjects from the GNC by using AI. The GNC, with its standardized MRI data of the spine of more than 30,000 people, has the potential to deliver further data toward standardized biometric reference values. To handle such large-scale big data and to extract normative morphometric values of the spine, AI tools are needed and have a great potential for quantitative MRI analyses [18] because manual analysis of such large data is not practical and reliable or would need tremendous expenditure in human resources [19].

In this manuscript, we will present a CNN-based software tool to analyze and generate morphometric MR imaging parameters of the entire spine of a large-scale MR dataset to generate reference values. CNN-based U-Nets are often used to segment pixels in images or imaging data [20]. In the beginning, we will briefly introduce the GNC and our AI approach. We will then present our results with a focus on statistical evaluation of the AI-based segmentation and finally end with the discussion.

## 2. Materials and Methods

### 2.1. German National Cohort 

All data used for this project was extracted from the ongoing prospective, multicentric GNC [21,22]. The participants (*n* = 11,254) enrolled in the GNC study were volunteers drawn from the general population. They underwent a whole-body MRI at 1 of 5 sites in Germany (Augsburg, Berlin, Essen, Mannheim, Neubrandenburg) during the years 2014 to 2016. The MR images were acquired on 3T clinical MRI scanners (Magnetom Skyra, Siemens Healthineers, Erlangen, Germany) with an identical configuration. A standardized examination protocol for the entire spine was used as described elsewhere [13]. The datasets were already combined to depict the entire spine by preprocessing using the manufacturer’s software. For further analysis, the pseudonymized DICOM datasets were converted into a NIFTI format. Segmentation was performed using ITK-Snap (Version 3.8.0) [23]. The demographic data for age, sex and height were provided by the GNC for every MRI data set.

### 2.2. Generation of Training Dataset

At first, volumetric analysis of the vertebral body and intervertebral disc, as well as the spinal canal, was needed as ground truth for the 3D-Convolutional Neural Network-based algorithm (3D-CNN). For the generation of this volumetric data, already acquired large-scale MRI datasets of the GNC were explored, consisting of T2-weighted sagittal images of the cervical, thoracal, and lumbar spine, which were composed during pre-processing by the manufacturer's software. MRI parameters were: echo time: 126 ms, repetition time: 4800 ms, slice thickness: 3.0 mm, field of view: 814 × 432 mm^2^.

For the analysis, 330 MRI data sets of representative subjects regarding the parameters of age, sex and height were selected from the GNC. Female and male subjects from all age (age subgroups: <30.5 y (years), 30.5–40.5 y, 40.5–50.5 y, 50.5–60.5 y, >60.5 y) and height groups (<165 cm, 166–175 cm, >175 cm) were randomly selected by an algorithm in an equal distribution. Then, segmentation and labeling of the spinal canal, the intervertebral discs and vertebral bodies of the 330 datasets were done manually by three readers. All segmented data was then reviewed in consensus by 2 board-certified radiologists, each with more than 10 years of experience in the evaluation of spinal MRI. Volumetric segmentations were done using the “smooth curve polygon mode” in ITK Snap (Version 3.8.0, [23]). Vertebral bodies were segmented from C2 to L5, but without consideration of vertebral arches, pedicles or the processes. The segmentation of the intervertebral discs was performed starting at the C1/2 level and including the L5/S1 level. The spinal canal was segmented along the dural sac from the foramen magnum to the level of the S1 upper plate. For quality control, 10% of all segmented datasets were randomly selected and reviewed by a board-certified neuroradiologist, ensuring a high-quality training dataset with the needed anatomical information for the machine learning algorithm.

### 2.3. Neural Network

For the neural network, we chose a 3D U-Net (Figure 1) similar to that defined by Çiçek et al. [24]. The basic concept of the U-Net is first to encode the MRI image on multiple resolutions so it can be decoded according to the desired output. As an encoder for each depth, we chose a convolutional layer to obtain the desired number of feature maps followed by a residual neural network (Figure 2) [25]. For the first encoder, the MRI image is the input, whereas for the remaining encoders, the output of the previous encoder, sub-sampled by factor 2, is the input. For the decoder, we chose a similar architecture to the encoder but reduced the number of residual layers from 2 to 1. The output of the encoders is used as input for the decoder. If available, the up-sampled output of the decoder of the lower depth is also used as the input for the decoder. So, the decoder can generate useful features taking into account the high resolution and local features from the encoder and the low resolution and global features from the previous decoders. The output of the last decoder is then the input of the last convolutional layer representing the output of the neural network.

We chose the depth of IV because additional depths did not lead to better performance. The same was observed for the number of residual layers in the encoders and decoders. We skipped batch norm layers because they resulted in more unstable training and did not lead to better results. Dropout also had a negative influence, probably because our large input augmentation preserved the neural network from overfitting.

The convolutional kernels at depth I have the kernel (3 × 3 × 3). Since the sagittal MRI has a pixel spacing of 0.9 mm and a slice thickness of 3.0 mm in depths II, III and IV of the U-Net, the convolutional layers only convolve in the sagittal plane with the kernel (1 × 3 × 3).

### 2.4. Training of the Deep Learning Algorithm

The segmented and labeled “ground truth” dataset was split into three parts, a training dataset (*n* = 250), a validation dataset (*n* = 50) and a test set (*n* = 30). While the training and the validation datasets are chosen randomly, the test set contains one sample from each cohort subgroup (5 age, 2 sex and 3 height groups). All datasets were non-overlapping. The AI algorithm was trained on 250 datasets, and the results were validated on the other 50 datasets. For training of the AI algorithm from the training set, sub-images of the size of 400 × 400 × 16 were sampled. The network is trained as a voxel-classifier using cross-entropy and focal loss (γ = 1.0). With 1.024 samples per epoch, the network trained 400 epochs using the ADAM optimizer [26] with a decreasing learning rate. To avoid overfitting on the training set, the MRI images are augmented with scaling, rotation, flipping, contrast changes and blurring in the sagittal plane (Table 1).

In the second step, the algorithm was tested and compared to human segmentation on the remaining 30 data sets. Therefore, we calculated a voxel-wise classification for all MRI images. The voxel classification was compared to the human segmentation by using precision, which is the fraction of relevant among the retrieved instances, the recall, which is the fraction of relevant instances that were retrieved and the Dice score, which characterizes the similarity of the 2 samples.

### 2.5. Extraction of Population-Based Data

Finally, the machine learning algorithm explored the sagittal spine MR-images of the full dataset (*n* = 11,254) regarding morphometry of vertebral bodies, intervertebral discs and spinal canal. During this process, 1039 data sets were rejected by the algorithm and had to be excluded for the following reasons: DICOM/NIFTI files were not readable, slice orientation was not orthogonal to each other, or there were some missing slices in the data set. For every data set, the anatomical dimensions of vertebral bodies and intervertebral discs, as well as the spinal canal, were labeled, resulting in an individual 3D model. The disc volume was calculated by the number of voxels multiplied by the voxel dimension. For the calculation of the paraxial spinal canal, the following steps were applied: a principal component analysis was conducted on the voxel coordinates classified as a specific disc. Both first eigenvectors were used to determine the disc plane. In the second step, all voxel coordinates classified as spinal canal were extracted, which are near the disc plane. A projection of these voxel coordinates on the disc plane resulted in a 2D point set. The area of the minimal convex hull around these points was determined as the spinal canal area.

Using the extracted parameters of disc volume and minimal paraxial area of the spinal canal at every spine segment from 10,215 datasets (median age: 53 years, max: 72 years, min: 20 years, distribution in the age subgroups: <30.5 years, *n* = 581; 30.5–40.5 y, *n* = 921; 40.5–50.5 y, *n* = 2705; 50.5–60.5 y, *n* = 3208; 60.5–80 y, *n* = 2800), morphometric standard values were calculated for age-, sex-, height-correlated groups. Afterward, the derived morphometric standard values were integrated into a supportive real-time analysis software tool.

## 3. Results

10.215 datasets of the GNC were successfully analyzed by using an AI-based machine learning algorithm with a 3D U-Net architecture. Vertebral bodies, intervertebral discs and the spinal canal could be segmented completely and automatically (Figure 3). The AI-based segmentations of the test datasets (*n* = 30) were statistically evaluated in comparison to the manual segmentations. The precision, recall and Dice score were all higher than 90% (Table 2).

The precision was 0.902 for intervertebral discs, 0.908 for vertebral bodies, and 0.926 for the spinal canal. The recall was 0.908 for the intervertebral discs, 0.909 for the vertebral bodies, and 0.924 for the spinal canal. It was possible to analyze the morphometry of the spine in excellent correlation to manually annotated data by trained radiologists (Figure 4). The Dice score, which is used to gauge AI model performance, ranges from 0 to 1. A Dice score of 1 corresponds to a pixel-perfect match between the AI model output segmentation and ground truth segmentation by the radiologists, and zero corresponds to no overlap between both segmentations. The Dice score was 0.905 for intervertebral discs, 0.908 for vertebral bodies and 0.925 for the spinal canal.

Based on the total of 10,215 datasets, it was possible to extract normative morphometric data for the spine regarding the vertebral body and intervertebral disc, as well as the spinal canal. By using this data, a machine learning-based software tool was developed to analyze the standardized GNC spine-MRI in real time and provide age-, sex-, and height-matched comparative data (Figure 5).

## 4. Discussion

Degenerative disc disease is a common imaging finding in the clinical routine. It may be present with or without symptoms, and it is usually detected when patients with symptoms like low back pain obtain diagnostic imaging. Although there are a variety of grading scales to categorize the imaging findings [14,27], little is known regarding to what extent DDD is physiological in a certain population, depending on their age and gender.

To answer the question if biometric measurements of the spine are physiological or pathological in a specific patient, MRI-based epidemiological studies are convenient for finding normative values. The number of participants necessary and the manpower needed to diagnose the imaging data to derive these normative values is huge. Manual evaluation and segmentation are not suitable approaches for this task. As mentioned before, in previous studies, the spine was just partially analyzed, or segmentation was performed just manually. In addition, the data of these studies were always acquired on a geographically local basis, like in the “Study of Health in Pomerania” [15,16]. In our study, for the first time, the morphometry of the entire spine of more than 11,000 representative subjects from all over Germany was evaluated by using AI.

Additionally, in recent years, the number of MRI examinations has already increased by over 71%, especially spine MRI in patients with chronic back pain. In contrast, there has been only growth of about 33% in radiologists within the last ten years. In the field of radiology, computer-aided tools with implemented AI algorithms constitute an exciting and promising field to compensate for radiologist staff shortage at some workflow stages and may help to provide support for diagnosis and treatment planning for intervention.

In our study, the AI approach was able to analyze morphometric features of the spine in a population-based study. The evaluated parameters were in excellent correlation to human analyses and at the same quality level in comparison to other U-net algorithms used in clinical studies [28,29]. Especially for specific MRI examinations, such as the analysis of spinal canal stenosis, our developed software tool could support physicians by analyzing the spine in real-time and providing age-, sex- and height-matched comparative data.

However, there are some limitations and different approaches to developing an even more accurate deep learning model. First, for the learning process of the AI, 3D augmentations during the training process and changes in the architecture with different shapes of the U-net could be used to test for higher achievable Dice coefficients. Second, the imaging parameters could be improved to optimize the segmentation results of the AI algorithm. Due to the slice thickness of about 3.0 mm provided by the GNC datasets, it is a great challenge for the algorithm to analyze the lateral edges of the intervertebral discs, vertebral bodies and spinal canal. An increased resolution and smaller slice thickness could improve the accuracy of the model.

Nevertheless, the standardized acquisition in large-scale MRI datasets, like the GNC data, is ideally suited for the approach with an AI-based deep-learning algorithm, but the question arises of how the AI algorithm would perform on not standardized clinical data. This evaluation of the AI-algorithm performance on data obtained in clinical routine must be done in future studies.

Another interesting approach in future studies could be the use of knowledge graphs, also known as a semantic network, which represents a network of different entities and makes a qualitative leap in knowledge representation [30,31]. When it comes to defining the normative morphometrical values of the spine, semantic information of different features like the volume and maximal or minimal height of VB could be evaluated by AI using knowledge graphs. This approach may allow analyzing the importance of different MRI features, e.g., variable image appearance or intensity ranges, regarding non-obvious correlations to reference values of the spine.

The extracted features and morphometric values are already the largest-scale database of the spine and enable capturing the variability of spine structures within a population of adults between 20 and 72 years. Although more than 11,000 MRI datasets were already evaluated in this study, the analysis of the additional 19,000 available datasets of the GNC could contribute to getting real normative values of the spine for the entire German population. Therefore, it is planned to analyze the additional datasets of the GNC and provide this data in the developed software tool with implemented normative data of the spine to other medical physicians and healthcare institutions. In addition, the robustness and reliability of our algorithm must be evaluated by using non-standardized acquired images from just parts of the spine and from different medical facilities and different devices. By using the developed software tool, MRI of the spine could be analyzed in real-time fully automatically, which supports the medical doctors by providing normative data. This could lead to better classification and understanding of, for example, degenerative changes of the spine like DDD.

We believe that the AI algorithm and the developed software will constitute an important breakthrough in spine imaging by adding normative data from the German population.

## 5. Conclusions

Using a CNN-based algorithm, the reliable segmentation of MRI datasets of the entire spine from the GNC, a population-based MRI study, was possible and achieved an excellent agreement with manually segmented datasets. With the analysis of the total GNC MRI dataset with almost 30,000 subjects, it will be possible to generate real normative standard values for different cohorts based on various demographic parameters in the future. By integrating these values in our software tool, medical doctors would be able to analyze MRI examinations of the spine in an easier way, extracting anatomical features through one-click quantitative measurements.

## Figures and Tables

**Figure 1 healthcare-10-02132-f001:**
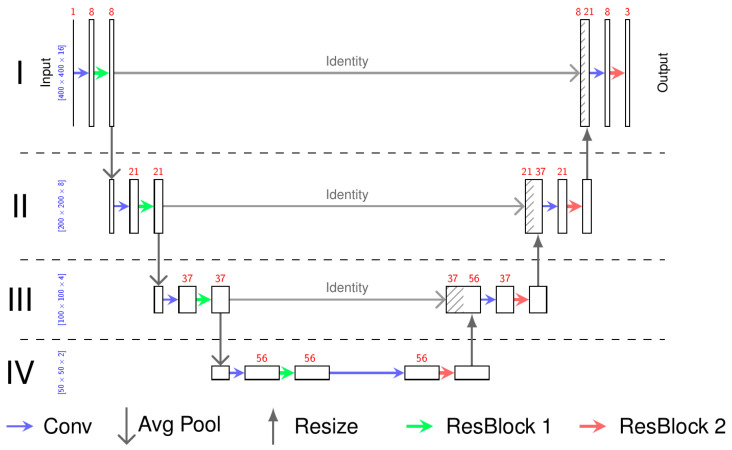
The architecture of the neural network: 3D U-Net. The input is a sample with the spatial dimension 400 × 400 × 16 of the MR image. The output has the same dimension containing the probability of the three classes (vertebral bodies, intervertebral discs, and spinal canal). The red numbers represent the number of feature maps, whereas the blue numbers define the spatial dimension of the feature maps. All arrows define functions between these feature maps as described in the picture. The Residual Blocks (ResBlock 1, ResBlock 2) are shown in Figure 2.

**Figure 2 healthcare-10-02132-f002:**
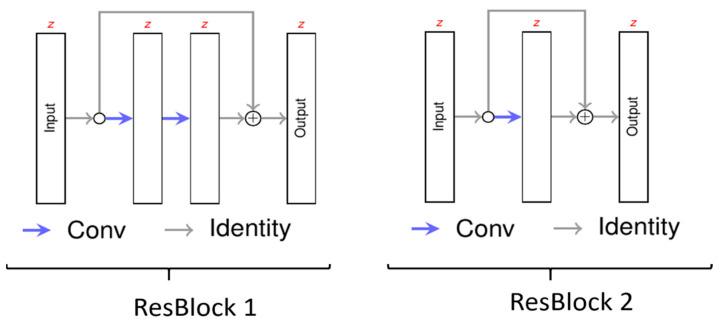
The architecture of residual blocks. Each layer has the same number of features. Each blue arrow (Conv) consists of a convolutional layer and a rectified linear activation.

**Figure 3 healthcare-10-02132-f003:**
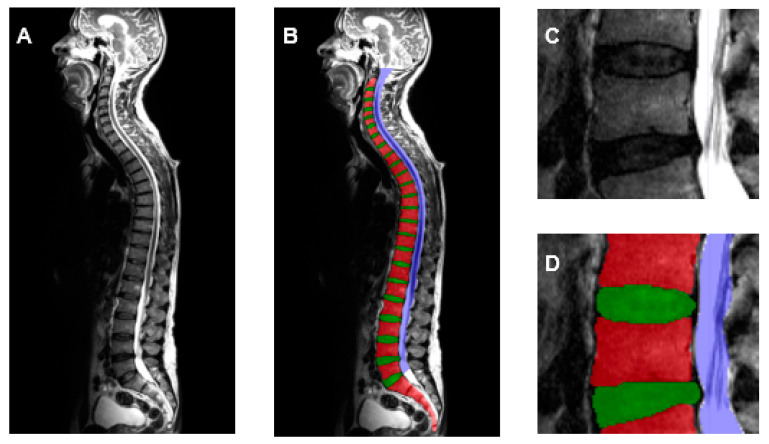
T2-weighted sagittal images of the entire spine (**A**) with segmentation by the AI algorithm (**B**) of the vertebral bodies (red), intervertebral discs (green) and spinal canal (blue). Enlarged view of the lumbar spine without (**C**) and after AI segmentation (**D**).

**Figure 4 healthcare-10-02132-f004:**
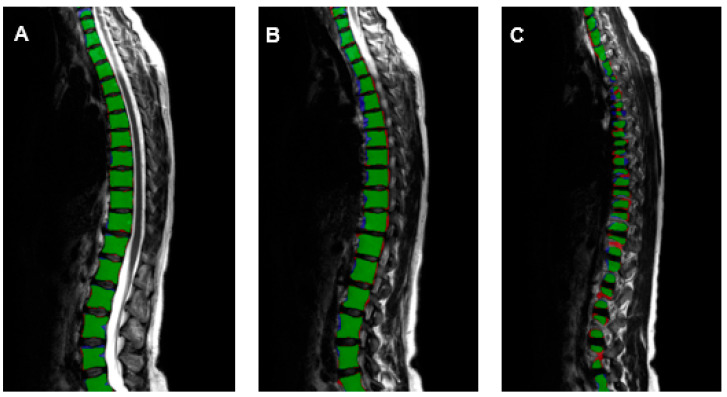
T2-weighted sagittal images (**A**)—medial, (**B**)—mediolateral, (**C**)—lateral of the entire spine with the evaluation of the AI-based vertebral body segmentation in comparison to the segmentation of the radiologists (green—true positive, blue—false negative, red—false positive).

**Figure 5 healthcare-10-02132-f005:**
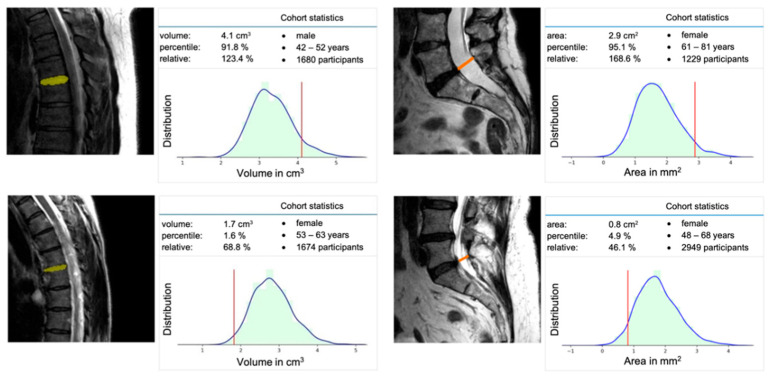
Screenshot from the software tool with quantitative evaluation of the intervertebral disc volume in correlation to the corresponding age and sex cohort (**left**). Quantitative evaluation of the minimal paraxial spinal canal area at the level of L5/S1 in correlation to the corresponding age and sex cohort (**right**).

**Table 1 healthcare-10-02132-t001:** Training parameters of the neural network.

Training Parameter	
sample size	400 × 400 × 16
optimizer	ADAM with a decaying learning rate
loss	cross-entropy with focal loss (γ = 1.0)
samples per epoch	1024
number of epochs	400

**Table 2 healthcare-10-02132-t002:** Statistical evaluation of the AI-based segmentation of the vertebral bodies (VB), intervertebral discs (VD) and spinal canal (SC) in comparison to the human segmentations.

	VB	VD	SC
Precision	0.908	0.902	0.926
Recall	0.909	0.908	0.924
Dice-score	0.908	0.905	0.925

## Data Availability

All supporting data is available from the GNC. Segmentation data is available from the corresponding author upon reasonable request.

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
