# Peer review of "Application of a Deep Learning Approach to Analyze Large-Scale MRI Data of the Spine"

_healthcare, 2022, doi:10.3390/healthcare10112132_

Round 1
Reviewer 1 Report
In this paper, the authors present a tool based in the use of convolutional neural network aimed to analyse spine-MRI.
After a detailed review, I consider that this is an interesting work. However, I consider that the authors need to expand their explanations, justifying and detailing their elections. For that reason, I propose to do a revision round.
Please, see my comments in the attached pdf file.

Reviewer 2 Report
- Introduction should add more discussion on the proposed approach.
- The literature review is missing. The process should be as follows:
i) Critical evaluation of the literature; ii) identifying the gap based on this critical evaluation of the literature; iii) proposing your hypothesis to address the identified gap; iv) posing the appropriate and relevant research question based on your proposed hypothesis; and finally explaining your proposed method.
- Evaluation metrics (Precision and Recall) were not described first,
- The discussion section should specify the importance of Knowledge Graphs as an important venue to accommodate MRI data. Authors can check the following works:
- Kou, Ziyi, et al. "HC-COVID: A Hierarchical Crowdsource Knowledge Graph Approach to Explainable COVID-19 Misinformation Detection." Proceedings of the ACM on Human-Computer Interaction 6.GROUP (2022): 1-25.
- Abu-Salih, Bilal. "Domain-specific knowledge graphs: A survey." Journal of Network and Computer Applications 185 (2021): 103076.
Round 2
Reviewer 1 Report
Thank you for the responses.
I do not have more questions.
Author Response
Dear Reviewer,
We would like to thank you once again for your favorable evaluation of our manuscript and the thoughtful comments.
We are very pleased that we could answer all of your questions.
Reviewer 2 Report
Authors have addressed my comments except the last one, authors can just discuss (in a paragraph) how knowledge graphs are important to be used in managing healthcare data such as MRI data. They can use the following literature:
- Kou, Ziyi, et al. "HC-COVID: A Hierarchical Crowdsource Knowledge Graph Approach to Explainable COVID-19 Misinformation Detection." Proceedings of the ACM on HumanComputer Interaction 6.GROUP (2022): 1-25.
- Abu-Salih, Bilal. "Domain-specific knowledge graphs: A survey." Journal of Network and Computer Applications 185 (2021): 103076.
